# NEURAL ATTENTION MEMORY

## ABSTRACT

Scaled dot-product attention has become the essence of state-of-the-art deep neural networks for various machine learning tasks. Though its ubiquitous accomplishments, it is inefficient for long sequence tasks and problematic for tasks requiring memory states such as compositional generalization. We propose a novel perspective of the attention mechanism by reinventing it as a memory architecture for neural networks, namely Neural Attention Memory (NAM). NAM follows the same query-key-value structure by constructing a memory matrix while reducing its computational complexity from quadratic to linear to the sequence length. NAM writes a memory matrix by adding outer products of value and unit key vectors, and reads it by multiplying the matrix with a unit query vector. We define read and write primitives of NAM and mathematically prove their functionalities. One benefit of NAM is that it can be a basis for efficient linear attention, namely normalized outer-product attention. We evaluate a NAM-based Transformer on long-range tasks and demonstrate NAM's efficiency and efficacy. Most importantly, NAM provides building blocks for memory-augmented neural networks. We propose two NAM-augmented neural networks, namely Long Short-Term Attention Memory (LSAM) and NAM Turing Machine (NAM-TM), and test their compositional generalization capabilities using four different tasks. LSAM replaces LSTM's long-term cell state with NAM memory matrix and NAM-TM implements a Turing tape structure using NAM read/write primitives. The experiments show that they have better computational power than Transformer and LSTM, as well as DNC. NAM opens up possibilities in diverse research problems, including hierarchical data modeling, efficient edge inference, and few-shot learning.

## 1 INTRODUCTION

Scaled dot-product attention (Vaswani et al., 2017) has become a core mechanism of state-of-the-art deep learning models for variety of machine learning tasks, including natural language processing (Devlin et al., 2018), multi-modal task (Li et al., 2019), and graph data processing (Hamilton et al., 2017). Specifically, the Transformers using the self-attention method have replaced recurrent neural networks (RNN) by outperforming them in most of the tasks. Despite its success, there exist limitations to the mechanism. First, it needs the information of the entire sequence to compute one attention so that its computational complexity becomes quadratic to the length of the sequence. Hence, it is inefficient for long sequence tasks (Tay et al., 2020) or edge inference environments (Tambe et al., 2020). Also, its stateless design enables efficient parallelism but makes it impossible to solve tasks that require memory states. Hence, Transformers fail to generalize the rules that require inductive bias (Dehghani et al., 2018) or compositional generalization (Lake & Baroni, 2018).

There have been studies designing neural networks with external memory to solve algorithmic tasks where Transformers fail. These memory-augmented neural networks (MANN) design differentiable read/write functions that can be trained by backpropagation. Some of them implement basic data structures like stack (Joulin & Mikolov, 2015) and queue (Grefenstette et al., 2015) while some implement complex memory structures using attention mechanisms (Graves et al., 2014; 2016). They outperform generic neural networks in synthetic algorithmic tasks but are considered impractical due to their complexities and inefficiencies.

In this work, we re-invent the attention mechanism as a memory architecture for neural networks, namely *neural attention memory* (NAM). NAM's design objective is to build simple, efficient, yet powerful external memory which also incorporates the attention mechanism. Following the same

query-key-value structure of attention, NAM stores key-value pairs to a *memory matrix* via additively writing their outer-products. Reading the memory matrix is simply done by multiplying the matrix with a unit query vector. We provide mathematical formulation for the read/write primitives, and make theoretical analyses showing that these read and write primitives can replace attention.

One big benefit of NAM is that it can perform attention in a more efficient way. By sacrificing the erasure capability of the NAM write operation, we can design an efficient and parallel attention mechanism, namely normalized outer-product attention. This special variant of NAM is almost equivalent to linear attention Katharopoulos et al. (2020), enjoying the same linear computational complexity to the sequence length. We evaluate NAM-based efficient Transformer in long-range arena (Tay et al., 2020) tasks. Its efficacy is on par with the base Transformer and Linear Transformer, implying that NAM can be an efficient alternative to the scaled dot-product attention.

The bigger value of NAM is that its read and write primitives can be building blocks for augmenting memory structures to deep neural networks. Using NAM read/write primitives, we design two memory-augmented neural networks (MANN), namely Long Short-term Attention Memory (LSAM) and NAM Turing Machine (NAM-TM). LSAM is a generic RNN architecture that replaces LSTM's long-term cell state with a memory matrix. Instead of additively writing a vector cell state, LSAM reads and writes the memory matrix using NAM primitives. The design combines strengths of attention and RNN while maintaining the same computational complexity as LSTM. NAM-TM is a MANN for algorithmic tasks, leveraging a Turing tape structure. A tape has read and write heads accessing the memory with NAM read/write primitives. They can move along the tape with four actions: NO-OP, LEFT, RIGHT, and JUMP. The actions are implemented as differentiable functions to enable end-to-end training with backpropagation.

We compare LSAM and NAM-TM to others in compositional generalization tasks of number sequence prediction (Nam et al., 2019), sequence reduction, and SCAN (Lake & Baroni, 2018). Specifically, we test their zero-shot generalization capability in length by training the models with sequences of limited length and validating them with longer sequences unobservable during training. The evaluation results show that their computational powers are superior to other baselines, including Universal Transformer (Dehghani et al., 2018) and DNC (Graves et al., 2016). While the generic LSAM model consistently outperforms the others, NAM-TM shows even better results at algorithmic tasks. The results indicate that NAM is a powerful method to implement memory in neural networks.

The efficient, simple, and flexible structure of NAM opens up new possibilities in multiple machine learning research fields. One straightforward application is leveraging NAM's efficiency for edge inference environment. Another possibility is using NAM for hierarchical data modeling by generalizing NAM with tensor products. Moreover, memorization of input-output mapping using NAM can be a solution for one-shot and few-shot learning.

The main contributions of this work are as follows:

- We re-invent the attention mechanism as a memory architecture for neural networks, namely neural attention memory (NAM).
- We present mathematical basis for NAM read/write primitives, and give theoretical proofs that NAM is equivalent to attention in certain conditions.
- We show that NAM can construct an efficient Transformer for long-range sequence tasks.
- We propose two memory-augmented neural network designs of LSAM and NAM-TM and show their capabilities in compositional generalization tasks.

## 2 BACKGROUND

### 2.1 SCALED DOT-PRODUCT ATTENTION

Attention mechanisms of deep neural networks (Bahdanau et al., 2014; Luong et al., 2015) provide differentiable methods of selectively attending items from a variable-length sequence. While there are multiple variations of attention mechanism, most of them share the same high-level structure: 1) compute the attention scores of the items, and 2) return the weighted sum of their vector representations using the scores. Among the variations, *scaled dot-product attention* (Vaswani et al., 2017) has been the most successful. For each token, there are a key vector and a value vector associated to it.

Given a query vector, the scores are determined by the scaled dot-product of the query and the keys. Then the output is computed by weighted sum of softmaxed scores and the value vectors.

The self-attention mechanism based on the scaled dot-product attention has proven to be very powerful, replacing the needs of RNNs. Since attention reaches every element of a sequence in an $O(1)$ path, it avoids the vanishing gradient problem (Hochreiter, 1998), enjoys high parallelism, and allows huge models with deeply stacked layers (Devlin et al., 2018; Brown et al., 2020). However, its stateless and parallel architecture also bring multiple limitations. First, the computational cost quadratically increases with the sequence length, making it inefficient for long-range contexts (Beltagy et al., 2020; Choromanski et al., 2020; Kitaev et al., 2020; Katharopoulos et al., 2020) and edge inference environments (Tambe et al., 2020). Also, the memory-less architecture lacks inductive bias (Dehghani et al., 2018), making it impossible to generalize inductive algorithmic rules (Kim et al., 2021). There are researches to resolve those issues, but a simple and practical solution is yet to be found.

## 2.2 MEMORY-AUGMENTED NEURAL NETWORK

In theory, RNNs are proven to be as powerful as Turing machines (Hyötyniemi, 1996). However, they fail to learn algorithmic patterns that require at least pushdown automata in practice (Nam et al., 2019). Therefore, there have been efforts to augment external memory architecture to neural networks, as known as memory-augmented neural networks (MANN) (Grefenstette et al., 2015; Joulin & Mikolov, 2015; Graves et al., 2014; 2016). The main challenge for MANNs is to design differentiable read and write functions that can be trained via back-propagation. Some MANNs use attention mechanism for differentiable read/write functions. For instance, Neural Turing Machine (NTM) (Graves et al., 2014) and Differentiable Neural Computer (DNC) (Graves et al., 2016) leverage attention mechanism for implementing content-based addressing. However, the addressing mechanisms are often not powerful enough so that they need to be augmented with complex extras, such as a link matrix with $O(N^2)$ cost. Hence, MANNs often become inefficient and complex so that they are considered impractical outside of the algorithmic task domain.

## 2.3 TRANSFORMER AND MEMORY

Transformers' attention mechanism can be understood as an external memory where all hidden activations are stored and selectively read. Since this method's storage footprint scales up with the sequence, there have been studies to limit the space complexity and scale to longer context by using explicit memory architecture to transformers. Transformer XL (Dai et al., 2019) has attempted to overcome the limited context length of Transformers by using segment-level recurrence and relative positional encoding. Compressive Transformer (Rae et al., 2019) and $\infty$-former (Martins et al., 2022) further have extended the idea by compressing the recurrent segment into a smaller memory and using less-precise unbounded continuous memory, respectively. While these works effectively address the scalability problem of Transformers, they have not extended their ideas to generic memory architecture that can be freely read and written.

# 3 NEURAL ATTENTION MEMORY

## 3.1 NAM READ AND WRITE PRIMITIVES

The main idea of NAM is implementing an attention mechanism via matrix-vector multiplication of a *memory matrix* $M \in \mathbb{R}^{d_v \times d_k}$, a *unit query vector* $q \in \mathbb{R}^{d_k}$ and a *read probability* $0 \le p_r \le 1$. Hereby $d_v$ and $d_k$ are feature dimensions of value and key vectors respectively. Then, the read operation ($RD$) of NAM computes the read vector $r \in \mathbb{R}^{d_v}$ as follows.

$$r = RD(M, q, p_r) = p_r M q$$

The memory matrix $M$ is written by adding *outer products* of unit key vectors $k_i$ and value vectors $v_i$. The write operation $WR$ updates the memory matrix with a write probability $p_w$ and an erase probability $p_e$ as below.

$$M' = WR(M, k, v, p_w, p_e) = M + p_w v k^\top - p_e M k k^\top$$

This write operation guarantees that reading with the same key vector yields the most recently written value. Such a property can be mathematically proven as below.

**Theorem 1.** *If $k$ is a unit vector and $M' = WR(M, k, v, 1, 1)$, $RD(M', k, 1) = v$.*

*Proof.* $M'k = Mk + vk^\top k - Mkk^\top k = v$ □

The verbal explanations of this theorem are as follows. First, $Mk = RD(M, k, 1)$ yields the value in associated to $k$ so that $-p_e Mkk^\top$ *erases* out the associated value from $M$. Then, adding $p_w vk^\top$ *writes* the new value on top of it.

### 3.2 Normalized Outer-product Attention

At each time step, the computational and space complexity of NAM are both $O(d_v d_k)$. Since $d_v$ and $d_k$ are per-head dimensions, it can be further reduced by the factor of the number of the heads $H$, to $O(H(d_v/H)(d_k/H)) = O(d_v d_k/H)$. In deep learning setups, these are identical to those of other neural network layers. On the other hand, scaled dot-product attention's compute and space complexities are $O(S(d_v + d_k))$, if the sequence has $S$ tokens. This opens up the opportunity for using NAM as an efficient linear attention mechanism for long-range sequential tasks or edge inference environments. Furthermore, the outer-product operation $vk^\top$ for NAM is often more efficient than dot-products in modern parallel HW since its compute complexity $O(d^2)$ is much higher than the amount of memory access required, $O(d)$.

However, the sequential nature of $WR$ can make NAM inefficient in parallel GPU and NPU architectures. This is because the erasure part $-p_e Mkk^\top$ of $WR$ depends on the current state of $M$. Hence, we can make $WR$ parallel by setting the erase probability $p_e$ to zeros. The reads can also become parallel if we make it *bidirectional* by reading the same final memory matrix $M = \sum_t p_w^{(t)} v^{(t)} k^{(t)\top}$, which is sum of outer product from every time step $t \in 1, \ldots, S$. Reading this memory matrix can act as an attention mechanism, namely *normalized outer-product attention*, as the following theorem holds.

**Theorem 2.** *If $k^{(1)}, \ldots, k^{(S)}$ are orthonormal, then $\forall i \in 1, \ldots, S, RD(M, k^{(i)}, 1) = p_w^{(i)} v^{(i)}$*

*Proof.* $Mk^{(i)} = \sum_t p_w^{(t)} v^{(t)} k^{(t)\top} k^{(i)} = p_w^{(i)} v^{(i)}$ as $k^{(t)\top} k^{(i)}$ is 1 when $t = i$ and 0 otherwise. □

In practice, such a hard guarantee is not possible since it is hard to make all the keys orthonormal. However, normalized outer-product attention can act as a soft attention mechanism if the keys are well separated. By removing the erasure process, it trades off the computational power of NAM for the maximal efficiency and parallelism. In the following sections, we show efficiency and efficacy of NAM as an attention mechanism in long-range sequential tasks (Section 4) and computational power of NAM in compositional generalization tasks (Section 5).

## 4 Efficient Transformer with NAM

### 4.1 Linear Transformer using NAM

Although NAM is theoretically capable of replacing scaled dot-product attention, there exists a danger of information loss due to the limited capacity of the memory $M$. Meanwhile, such an information loss is not an issue for Transformers because they utilize hidden activations from the entire sequence. Hence we need empirical evidences that NAM is an effective alternative of attention. Hereby we implement an efficient linear bidirectional transformer using normalized outer-product attention, namely *NAM-Transformer*. As a proof-of-concept, we prove the efficacy and efficiency of NAM by evaluating NAM-Transformer at long-range sequence tasks.

NAM-Transformer is the most basic way of leveraging normalized outer-product attention, by setting the read/write probabilities to 1. Given queries $Q \in \mathbb{R}^{S \times d_k}$, keys $K \in \mathbb{R}^{S \times d_k}$, and values $V \in \mathbb{R}^{S \times d_v}$ of a sequence with $S$ tokens, NAM-Transformer's self-attention is computed as follows.

$$SelfAttn_{NAM}(Q, K, V) = (V^\top \mu(K))\mu(Q)$$

Hereby $\mu$ is a unit-vector normalization function applied to each row vector of $K$. Computationally, this is almost identical to the self-attention of Linear Transformer (Katharopoulos et al., 2020). The

differences from Linear Transformer to ours are: 1) we use unit vector normalization instead of ELU kernel function, 2) so that we do not need to compute the causal masking factor $Z$ (Katharopoulos et al., 2020). That is, one can think Linear Transformer as a special variant of NAM, sacrificing some computational capabilities for parallelism and efficiency.

## 4.2 LONG-RANGE TASK EVALUATION

Table 1: Accuracy (%) and relative training speedup comparison of the three Transformer models in listops (Listops), text classification (Text), pixel-level image classification (Image) tasks.

| | Listops | | Text | | Image | |
|---|---|---|---|---|---|---|
| Model | Acc | Speedup | Acc | Speedup | Acc | Speedup |
| Transformer | 29.75 | 1 | 57.28 | 1 | 40.16 | 1 |
| Linear Transformer | 36.5 | 2.59 | 64.27 | 2.35 | 38.76 | 10.02 |
| NAM-Transformer (Ours) | 36.1 | 2.66 | 63.49 | 2.44 | 37.18 | 10.07 |

We test our NAM-Transformer on long-range arena tasks (Tay et al., 2020) that have been used to compare efficient Transformer architectures. Since the previous work has already proven that Linear Transformer is as capable as others, we conduct partial comparison of ours to the original Transformer and Linear Transformer. Table 1 shows that NAM can be an efficient alternative to Transformers. Despite the information loss and missing normalizing factor $Z$, the task accuracy of NAM is very similar to the others, even surpassing the original Transformer in some cases. We see bigger speedups at the image classification task because the models have much smaller per-head dimension than others (16 vs 64). Recall that the computational complexities of both linear transformer and NAM transformer are quadratic to the per-head dimensions. Hence, the image classification result shows that smaller per-head dimension can bring bigger speedups for NAM, but may result in lesser capacity. Overall, the normalized outer-product attention is as powerful as the scaled dot-product attention while enjoying greater computational efficiency. In other words, NAM can be an efficient and effective alternative for attention mechanism.

**Setup** We use the code base of the original benchmark [1] and implemented our NAM encoder by modifying the Linear Transformer implementation. The three models share the same hyperparameters and the only differences come from the attention algorithms. The evaluations are run on the Ubuntu 20.04 system with RTX 3080. The results differ from the original work because we used smaller batch sizes due to the limited VRAM capacity. All other experimental setups, including hyperparameters, are identical to the original benchmark (Tay et al., 2020). Details are available in the source code included as a supplementary material.

## 5 NAM AUGMENTED NEURAL NETWORKS

In this section, we design two types of memory-augmented neural networks (MANN) based on NAM, namely *Long Short-term Attention Memory* (LSAM) and *NAM Turing Machine* (NAM-TM). LSAM is a generic recurrent neural network architecture derived from LSTM (Hochreiter & Schmidhuber, 1997). LSAM replaces the long-term cell state of LSTM with a NAM memory matrix. NAM Turing Machine is a MANN design for algorithmic tasks inspired by Neural Turing Machine (Graves et al., 2014). Its read and write heads can move along the tape with four actions: left, right, no-op, and jump. Implementations of the heads are based on NAM read/write primitives. We evaluate the models with tasks of number sequence prediction, sequence reduction, and SCAN. We test their compositional generalization capability by splitting the data based on the sequence lengths.

## 5.1 LONG SHORT-TERM ATTENTION MEMORY

Long Short-term Memory (LSTM) (Hochreiter & Schmidhuber, 1997) leverages two recurrent state vectors: the short-term hidden state and the long-term cell state. To mitigate the problems

---

[1]github.com/google-research/long-range-arena

of vanishing/exploding gradients, the cell state is additively updated using forget and input gates. Then, the output gate selectively reads the cell state to determine the hidden state. Long Short-term Attention Memory (LSAM) follows the same principle. Instead of using the vector cell state, it leverages the memory matrix $M_t \in \mathbb{R}^{d^2}$ which is also additively updated using the NAM write primitive. The hidden state $h_t \in \mathbb{R}^d$ is retrieved by reading the memory matrix. Given the input $x_t \in \mathbb{R}^d$, the update rule $M_t, h_t = LSAM(x_t, M_{t-1}, h_{t-1})$ is defined as follows.

$$[q_t : k_t : v_t] = W_{qkv}[x_t : h_{t-1}] + b_{qkv}$$
$$< p_r, p_w > = \sigma(W_{rw}[x_t : h_{t-1}] + b_{rw})$$
$$M_t = WR(M_{t-1}, \mu(k_t), v_t, p_w, p_w)$$
$$h_t = RD(M_t, \mu(q_t^i), p_r)$$

Hereby : is the concatenate operator, $\sigma(.)$ is the sigmoid function, and $W_{qkv}, W_{rw}, b_{qkv}, b_{rw}$ are trainable weights and biases. Although the memory matrix $M_t$ has much higher capacity than a vector cell state, the computational complexity of LSAM is identical to that of LSTM. This is because $WR$ and $RD$ have complexity of $O(d^2)$ which is identical to that of matrix-vector multiplication of the weights and the states. Like Transformers, we can design multi-headed LSAM by concatenating per-head states $M_t^i$ and $h_t^i$. Bidirectional LSAM is also possible by splitting the multiple heads into two directions. The backward heads are updated in the opposite direction by the rule of $M_t^j, h_t^j = LSAM(x_t, M_{t+1}^j, h_{t+1}^j)$.

The LSAM architecture combines the strengths of recurrent neural networks (RNN) and Transformers. Since LSAM follows the RNN design so that it enjoys strengths of RNNs such as low computational cost for inference and recurrent inductive bias. Additionally, it benefits from the strengths of attention because reading and writing the memory matrix natively incorporates the attention mechanism.

## 5.2 NAM TURING MACHINE

Neural Turing Machine (NTM) (Graves et al., 2014) is one of the early neural networks that implement external memory structure with differentiable read and write methods. It is a basis of Differentiable Neural Computer (DNC) (Graves et al., 2016) which has proven to be effective at solving a variety of algorithmic tasks such as answering synthetic questions and finding shortest paths. Their external memory matrix is accessed by read and write heads using differentiable attention mechanisms.

Hereby we design NAM Turing Machine (NAM-TM) which adopts the design principles of NTM and DNC. The main idea of NAM-TM is to treat the tape state $T = [v_1, v_2, ...v_n, \mathbf{0}, ...] \in \mathbb{R}^{L \times d}$ as a memory matrix as follows.

$$T = [v_1, v_2, ...v_n, \mathbf{0}, ...] = \sum_i v_i e_i^\top \qquad (v_i \in \mathbb{R}^d, e_i \in \mathbb{R}^L)$$

Hereby $e_i$ are standard basis vectors $< 0, 0, ...1, ..., 0 >$ of $\mathbb{R}^L$ where $L$ is the size of the tape. This tape state can now be accessed by using NAM read/write primitives.

NAM-TM is a differentiable function that takes the tape state $T$, read and write heads $H_r, H_w \in \mathbb{R}^L$, and the input vector $x \in \mathbb{R}^d$ and produces the read output $R \in \mathbb{R}^d$ along with the updated tape and head states $T', H_r', H_w'$.

$$R, T', H_r', H_w' = NAMTM(T, H_r, H_w, x)$$

The read and write heads are positional vectors to attend the memory matrix for reading and writing the states. At each time step, the positions can be updated by four actions: LEFT, RIGHT, NO-OP, and JUMP. They are controlled by a controller neural network $nn\_control(x)$ which emits read and write probabilities $p_r, p_w$, action probabilities $p_{right}, p_{left}, p_{noop}, p_{jump}$ for each head and a unit jump query vector $q_{jump} \in \mathbb{R}^d$. Given the controller outputs and the value $v = W_v x$ to write, reading and writing the memory are conducted by NAM primitives as follows.

$$R = RD(T, H_r, p_r)$$
$$T' = WR(T, H_w, v, p_w, p_w)$$

Then, each head is updated to the next position based on the action probabilities. LEFT and RIGHT actions can be performed by the differentiable $roll$ function, which is a linear transformation $\mathbb{R}^L \longrightarrow \mathbb{R}^L$ mapping $e_i$ to $e_{i+1}$.

$$(H_{LEFT}, H_{RIGHT}) = (roll^{-1}(H), roll(H))$$

The jump position $H_{JUMP}$ is determined by reading the transpose of a key tape $K$ with the unit query vector $q_{jump}$. The key tape is written in the same way as the tape $T$, but it stores the corresponding unit key vector derived using the weight matrix $W_k \in \mathbb{R}^{d \times d}$ and the unit vector normalization $\mu(.)$.

$$K' = WR(K, H_w, \mu(W_k x), p_w, p_w)$$
$$H_{JUMP} = RD(K'^{\top}, q_{jump}, 1)$$

One can understand the transpose of the key tape $K^{\top} = \sum_i e_i k_i^{\top}$ as a jump table. Reading $K^{\top}$ with a key vector $k_i$ returns the corresponding position vector $e_i$ if the keys are orthonormal. Finally, the next head position $H'$ is updated as a weighted sum of the positions.

$$H' = p_{noop} \times H + p_{left} \times H_{LEFT} + p_{right} \times H_{RIGHT} + p_{jump} \times H_{JUMP}$$

While all of the computations are technically done with a fixed tape length $L$, none of the trainable parameters depend on the value of $L$. That is, a NAM-TM trained on certain length $L$ can be applied to any tape length $L'$ without modification nor re-training. Theoretically, it can be extended to infinite-dimensional Hilbert spaces.

There are multiple strengths in NAM-TM design compared to the memory structures of NTM and DNC. First, unlike NTM and DNC, NAM-TM's building blocks are simple and computationally efficient. Second, NAM-TM's addressing mechanism is based on the query-key-value attention mechanism of NAM, which is more powerful than the content-based attention mechanism used in NTM and DNC. Finally, NAM-TM's design is flexible in that it is easy to add/remove transition rules of the read/write heads. For example, the JUMP transition rule is optional in that a Turing machine only requires LEFT and RIGHT transitions in theory. One can also add another transition rule for the head positions if the rule can be defined with differentiable functions.

## 5.3 COMPOSITIONAL GENERALIZATION TASKS

We test the computational powers of LSAM and NAM-TM using three types of algorithmic tasks in compositional generalization setups. First, number sequence prediction (Nam et al., 2019) task (NSP) is a suite of synthetic problems to predict the following digits of the numerical sequences. It can test the compositional generalization capability by testing/validating the models with the longer decimal numbers that are never observed during the training stage. In this setup, many models often suffer from drastic fall of test/validation accuracy, due to lack of inductive bias (Kim et al., 2021). We use two representative sequences from NSP: Fibonacci (Fib) and Palindrome (Palin). The two tasks require the generalization of digital addition and sequence reversal rules, respectively.

Next is a sequence reduction (Reduce) task with a simple rule: given the sequence of digits, the target is the reduced sequence by skipping zeros. This is a task that can be easily solved by a proper Turing machine. We also use a similar compositional generalization setup by testing/validating using the longer target sequences that are longer than any target sequences in the training dataset.

For NSP and Reduce tasks, we use training datasets with $d = 1 \dots 10$ decimal digit sequences in little-endian order. Then we validate/test the models with two validation sets (ID, OD-easy) and a test set (OD-hard) for each task. An in-distribution (ID) validation set consists of $d = 5 \dots 10$-digit sequences, and an out-of-distribution validation set (OD-easy) consists of $d = 11 \dots 13$-digit sequences. The harder out-of-distribution test set (OD-hard) consists of $d = 14 \dots 16$-digit sequences, challenging the models with generalization to longer contexts. A training dataset has 25600 samples, and each validation/test set has 2048 samples.

The last task is SCAN (Lake & Baroni, 2018) task which has familiarized the concept of compositional zero-shot generalization. SCAN consists of simplified natural language input sequences and the

corresponding output action sequences. Among many data split methods, we use train/test split by length [2] to be consistent with the other tasks. Since SCAN only has a limited number of sequences, we only have two datasets of train and test.

| Fibonacci | Palindrome | Reduce |
|---|---|---|
| Input:  0 9 _ 7 3 9 / * * * * * | 8 4 5 3 1 / * * * * * * | 2 0 6 0 9 1 / * * * * * |
| Target: * * * * * * / 7 2 0 1 | * * * * * / 1 3 5 4 8 | * * * * * * / 2 6 9 1 |

Figure 1: Input and output sequence examples of the tasks. The Fibonacci sequence is given in the little-endian order.

We format the problems as masked sequence completion problems as shown in Figure 1. An input sequence consists of input tokens followed by masks, and an output sequence consists of target tokens following the masks. Since the input and output sequences are 1:1 matched, we can compare to baseline models without sequence-to-sequence structures like DNC. We choose four baseline models to compare: a bidirectional Transformer encoder (TF) (Devlin et al., 2018), an 2-layer LSTM model with attention (Bahdanau et al., 2014), an Universal Transformer (Dehghani et al., 2018), and a Differentiable Neural Computer (DNC). The TF model follows the architecture of BERT$_{medium}$ and the hyperparameters of the other models are adjusted to have similar parameter counts. Our LSAM and NAM-TM networks have two LSAM layers and two NAM-TM layers respectively, whose hyperparameters are also adjusted to have similar parameter counts. As an ablation study, we also evaluate NAM-TM design without the JUMP transition (No Jmp). All experiments are run with PyTorch 1.10 on the system with Ubuntu 20.04 and RTX 3080. Each experiment takes less than five hours to run 200 training epochs. The implementation details and hyperparameters can be found at the source code in the supplementary materials.

## 5.4 EVALUATION RESULT

Table 2: Sequence accuracy (%) comparison on the compositional generalization tasks.

| | Model | TF | LSTM | UT | DNC | LSAM | NAM-TM | No Jmp |
|---|---|---|---|---|---|---|---|---|
| | Parameters | 28.7M | 31.5M | 26.0M | 40.2M | 25.8M | 23.3M | 22.2M |
| Palin | ID | 100% | 100% | 100% | 100% | 100% | 100% | 100% |
| | OD-easy | 65.6% | 100% | 99.2% | 100% | 100% | 100% | 100% |
| | OD-hard | 19.0% | 100% | 5.2% | 100% | 100% | 100% | 100% |
| Fib | ID | 98.9% | 46.2% | 99.1% | 46.9% | 100% | 97.4% | 40.4% |
| | OD-easy | 0.4% | 8.3% | 21.3% | 1.8% | 39.9% | **89.7%** | 19.8% |
| | OD-hard | 0.0% | 0.0% | 0.1% | 0.0% | 2.9% | **71.5%** | 1.1% |
| Reduce | ID | 99.6% | 99.5% | 98.7% | 100% | 99.9% | 100% | 100% |
| | OD-easy | 0.2% | 91.5% | 15.7% | 95.7% | 96.8% | **100%** | **100%** |
| | OD-hard | 0.0% | 63.6% | 0.0% | 60.1% | 77.6% | **100%** | **100%** |
| SCAN | Train | 99.8% | 99.9% | 99.6% | 99.3% | 99.5% | 99.6% | 99.9% |
| | Test | 0.0 | 14.5% | 4.5% | 8.6% | **14.9%** | 11.2% | 9.6% |

Table 2 shows the sequence accuracy comparison of the models in compositional generalization tasks. We report the sequence accuracies of the models, where a wrong prediction of one token is counted as a failed prediction of the entire sequence. To avoid cherry-picking, the epochs of the best OD-easy validation accuracies are presented for the evaluation results. However, we present the results of the best test accuracy for SCAN tasks because there are only two available datasets (Train, Test).

Although LSAM's architecture is not specifically designed for algorithmic tasks, LSAM performs consistently better than the other baselines. Surprisingly, it performs better than DNC, which is a MANN model designed for such algorithmic problems. This is not a matter of model capacity

---

[2]github.com/brendenlake/SCAN

because LSAM has a slightly smaller parameter count. The results imply that the computational power of NAM memory architecture is superior to that of DNC's memory architecture.

As expected, NAM-TM performs significantly better in algorithmic tasks (Palin, Fib, and Reduce) possibly due to its specialized architecture. Especially, it finds easier to solve OD-hard problems, whereas the other models experience steep performance decline. A potential explanation is that the action-based positional transitions provide robustness in long-context cases. However, NAM-TM is not the best-performing model for the SCAN task.

NAM-TM remains effective at palindrome and reduction tasks even without the JUMP transition (No Jmp). Hence, LEFT and RIGHT transitions seem to be enough for emulating simple Turing machines, but more complex transition rules can augment the computational power of NAM-TM. This suggests extending NAM-TM to variety of tasks by augmenting specialized transition actions.

## 6    FURTHER APPLICATIONS

Since NAM offers simple and flexible building blocks for memory architecture, there are infinite number of potential applications. Hereby we suggest a few possible extensions of NAM. While we do not evaluate the ideas in this work, we leave them as future work.

**Hierarchical data modeling**    Tensor product, the key operation behind NAM, is not limited to two-dimensional outer product. Any dimensional tensors $T_i$ can construct a memory tensor $M$ by performing sum of tensor products $M = \sum_i T_i \otimes k_i$, with the unit key vectors $k_i$. For example, a document tensor $D$ can be constructed by a sum of tensor products of sentence-level keys and sentence matrices $S_i$, each of which is also a sum of outer products of word embeddings and word-level keys. *Nested attention* to such a hierarchical memory can be conducted by two inner products with a sentence-level unit query vector $q_s$ and a word-level unit query vector $q_w$.

**Efficient edge inference**    While Transformers are very successful in many ML tasks, deploying such models for edge inference has been a challenging task since Transformer's computation and memory cost per each time step varies with the sequence length. However, NAM's cost does not depend on the sequence length at all. Also, the outer product operation is more compute intensive than the dot products, making NAM more friendly to high-throughput accelerators. Hence, NAM can be an efficient Transformer alternative for edge inference.

**Few-shot learning**    Given a input-output vector pair $(x, y)$, it is possible to conduct *one-shot learning* of a weight matrix $W$ by $WR(W, y/|x|, x/|x|, 1, 1)$. This is because $Wx$ is guaranteed to return $y$ by Theorem 1. Therefore, we can implement one-shot or few-shot learning as memorization to NAM. This aligns well with the human behavior since we conduct few-shot learning by memorizing, not by repetitive training.

## 7    CONCLUSION

We proposed a redesign of attention mechanism to construct a differentiable memory for neural networks, namely neural attention memory (NAM). Following the same query-key-value structure of scaled dot-product attention, NAM first writes the memory matrix by adding outer products of key-value pairs. Then we can read it by multiplying the matrix with a unit query vector. The first strength of NAM is that its computational complexity does not rely on the sequence length. The long-range arena evaluations showed that NAM based attention, namely normalized outer-product attention, is an efficient and effective alternative for scaled dot-product attention. Next, NAM can be a powerful basis for constructing MANN models. We designed two NAM-based MANNs: LSAM for generic sequential tasks and NAM-TM for algorithmic tasks. In compositional generalization tasks, both outperformed other baselines such as Universal Transformer and DNC, indicating that NAM is a more powerful mechanism for implementing memory in DNNs. Finally, it opens up further research possibilities in the fields of hierarchical data modeling, edge inference, and few-shot learning.

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
