# OpenReview forum: "Neural Attention Memory"
_ICLR.cc/2023/Conference — Submitted to ICLR 2023_

### Official Review · Reviewer_g5E5 · 2022-10-18

**Confidence:** 3
**Correctness:** 3
**Technical Novelty And Significance:** 3
**Empirical Novelty And Significance:** 2
**Recommendation:** 6

**Clarity, Quality, Novelty And Reproducibility:**

The motivation is clear and the two sets of experiments are adequate to show how NAM addresses the limitations of current systems. However, I still think that you could rewrite some parts in order to make the connection between $\S3$ and $\S4$ easier.

The paper is well structured and the idea is interesting. I think, however, that you could still compare your methods to other works (e.g., other long-range arena transformers and more recent MANN models).

To the best of my knowledge, NAM’s formulation is novel. In practice, a transformer with NAM is similar to a Linear Transformer with some differences, highlighted in $\S4.2$ (unit vector normalization instead of ELU kernel function; no need to compute the causal masking factor Z).

Code is provided as supplementary material. Are you planning to release the code to everyone?



**Strength And Weaknesses:**

Strengths:

* The paper is well written and is easy to follow.
* The motivation is very clear from the beginning: trying to address two limitations of current attention mechanisms (inefficiency when dealing with long sequences and stateless design).

Weaknesses:
* The discussion about memory-augmented neural networks (MANN) in $\S2.2$. seems limited. It would be beneficial to discuss the similarities/differences of your proposal and recent works that try to augment the transformer architecture with a memory network (e.g., [1], [2]). The same applies for the experiments in $\S5$.
* The writing of $\S4.1$ could be improved. In particular, I find the explanation of NAM-based attention a bit confusing and unstructured. Also, I don’t think the connection between $\S3$ and $\S4$ is very clear.
* Several transformer architectures that try to efficiently model long sequences have been proposed in the last couple of years. I believe that your experimental results would be stronger if you compared NAM to some of these – the linear transformer is just one.

Minor Comments/Questions:

* I would slightly change the sentence “Attention mechanisms of deep neural networks (Bahdanau et al., 2014; Luong et al., 2015) provide differentiable methods of choosing (attending) one item from a variable-length sequence.”, because attention may be spread/not concentrated on one item only.
* Typo in “algorithmic patterns that require more that pushdown automata in practice”
* Is something missing in “Note that $V^\top(p_w \odot \mu(K))$ is one of the special cases of constructing the memory matrix by setting the erase probabilities to zeros W R operations.”?
* Typo (*hyperparameters*) in “The implementation details and hypereparameters can be found at the source code in the supplementary materials.” Also, it would be better to have the hyperparameters listed in, e.g., an appendix, and not only as part of the code.
* Can you further comment on “However, NAM-TM is not the best-performing model for the SCAN task.”?
* Typo in “For example, a document tensor $D$ can constructed by sum”
* Typo in “NAM’s cost does not depend the sequence



[1] Compressive Transformers for Long-Range Sequence Modelling, Rae et al., ICLR 2020

[2] $\infty$-former: Infinite Memory Transformer, Martins et al., ACL 2022



**Summary Of The Paper:**

This paper proposes neural attention memory (NAM), an alternative to the standard attention mechanisms, which suffer from well known limitations: their quadratic dependency with respect to the sequence length, and thus their inefficiency when dealing with long sequences, as well as their difficulty to solve tasks that require memory states.

NAM follows the same query-key-value structure of scaled dot-product attention; it first writes a memory matrix by adding outer products of key-value pairs, and then reads it by multiplying the memory matrix with a unit query vector.  NAM attention reduces the computational complexity to linear with respect to the sequence length.

They perform experiments on long-range arena tasks, comparing a transformer with NAM self-attention with a vanilla transformer and with the Linear Transformer of Katharopoulos et al. (2020). They further experiment with their memory augmented neural network designs (LSAM and NAM-TM) on compositional generalization tasks like number sequence prediction, sequence reduction, and SCAN.


**Summary Of The Review:**

The paper is well written and addresses an important problem. The formulation is interesting and might be used for other applications that were not explored in this paper. See the Weakness section for further details on what I think the paper would benefit from.

---

> ### Author Response · Authors · 2022-11-07
> **Response to Reviewer g5E5**
>
> First of all, we appreciate your comments and feedbacks. Your comments about writings and typos are very helpful and we will reﬂect them for the revision of the paper.
>
> > The discussion about memory-augmented neural networks (MANN) in §§2.2. seems limited. It would be beneficial to discuss the similarities/differences of your proposal and recent works that try to augment the transformer architecture with a memory network (e.g., [1], [2]). The same applies for the experiments in §§5.
>
> Thank you for suggesting two papers to read (Rae et al 2020, Martins et al 2022). We will add them to the discussions of Section 2 in the revision of the paper.
>
> > The writing of §§4.1 could be improved. In particular, I find the explanation of NAM-based attention a bit confusing and unstructured. Also, I don’t think the connection between §§3 and §§4 is very clear.
>
> As mentioned in the answers to the common comments, we agree that Section 4 is confusing to readers. The revision will focus on better writings in Section 3 and 4.
>
> > Several transformer architectures that try to efficiently model long sequences have been proposed in the last couple of years. I believe that your experimental results would be stronger if you compared NAM to some of these – the linear transformer is just one.
>
> Please see the answers to the common comments. Since Section 4 is a proof-of-concept, the main objective of the evaluation is to show that NAM is on-par with Linear Transformer. We will add more data points to Table 1 for better analysis.
>
>
> > Code is provided as supplementary material. Are you planning to release the code to everyone?
>
> The code is yet private but will be open to the public after the paper is published.

---

### Official Review · Reviewer_Ln3M · 2022-10-22

**Confidence:** 4
**Correctness:** 4
**Technical Novelty And Significance:** 3
**Empirical Novelty And Significance:** 3
**Recommendation:** 6

**Clarity, Quality, Novelty And Reproducibility:**

The paper's clarity is great: I managed to follow the details seamlessly.

The quality is also good: the experiments are well-executed.

Originality gets close to existing ideas (Linear Transformer) but it is nevertheless unique in its implementation and applications on relevant tasks.



**Strength And Weaknesses:**

Strengths:

1. Simple and working idea.

2. Meaningful implementation that can be applied in hardware efficiently, and it could help few-shot learning as well.

Weaknesses:

1. The gains are only marginal compared to the Linear Transformer (see Table 1).

2. It seems to me that the Self-attention NAM model is a bit disconnected from Attention NAM. Namely, currently posed, there are no read/ write operations in SelfAttn_NAM. Could you elaborate a bit further, or potentially test some SelfAttn mechanisms with read / write?

3. It would be nice to see experiments on the domains you suggested trying, e.g. few-shot learning.

4. Minor:

* I think you mean "bigger speadups at image classification" in place of  "bigger speadups at text classification" when discussing Table 1.

**Summary Of The Paper:**

The paper proposes neural attention memory (NAM) as an efficient memory-augmented attention mechanism. The authors propose a well-motivated read and write mechanism that is efficient and convenient for hardware implementations. Then, they show a self-attention mechanism inspired by NAM. The resulting NAM Transformer is comparable to Linear Transformer (direct competitor). The authors also design an LSTM-like and an NTM-like architectures that use NAM as attention mechanisms. The authors demonstrate that the NAM-augmented architectures are superior to direct competitors, such as LSTM, NTM, DCN, UT and TF on simple algorithmic tasks, such as Fibonnaci, Palindrom, Reduce and SCAN. Sometimes the proposed methods generalize very well on OOD tasks.

**Summary Of The Review:**

The paper is well executed and the constructions are meaningful, so I recommend weakly the paper to be accepted. To improve the paper: it would be nice to experiment with read and write into the self-attention mechanism. Further improvements might come from trying some simple few-shot learning tasks.

---

> ### Author Response · Authors · 2022-11-07
> **Response to Reviewer Ln3M**
>
> First of all, we appreciate your comments and feedbacks. We are actively working on revising the paper reﬂecting your comments.
>
> > 1. The gains are only marginal compared to the Linear Transformer (see Table 1).
>
> > 2. It seems to me that the Self-attention NAM model is a bit disconnected from Attention NAM. Namely, currently posed, there are no read/ write operations in SelfAttn_NAM. Could you elaborate a bit further, or potentially test some SelfAttn mechanisms with read / write?
>
> Please see the answers to the common comments. We understand the explanations of NAM-Transformer are confusing and are working on the better writing for the revision.
>
> > 3. It would be nice to see experiments on the domains you suggested trying, e.g. few-shot learning.
>
> We are working on the one-shot/few-shot learning experiments using NAM. We cannot add the results to the Nov 18 revision, but trying to add them later when we have enough results to present.

---

### Official Review · Reviewer_ek6u · 2022-10-25

**Confidence:** 3
**Correctness:** 3
**Technical Novelty And Significance:** 2
**Empirical Novelty And Significance:** 2
**Recommendation:** 3

**Clarity, Quality, Novelty And Reproducibility:**

The experiments in section 5.4 lack detail. For example, the SCAN task includes four different splits: simple, length, jump, and turn left. It's unclear on which split the author tested their model. The novelty of this paper is also limited, a similar technique has been proposed by the linear transformer.

**Strength And Weaknesses:**

Strength:
1. The proposed NAM transformer is fairly efficient compared to the standard transformer model and slightly faster than the linear transformer model.
2. The newly proposed LSAM and NAM TM models expand the memory capacity of the original LSTM and NTM model and achieve strong compositional generalization.

Weakness
1. The NAM is not substantially different from the linear transformer and achieves lower performance compare to the linear transformer while the speedup is similar.
2. The performance of LSAM and NAM TM still falls behind the SOTA models in the SCAN task.

**Summary Of The Paper:**

The paper proposes a memory-based attention mechanism. It leverages the outer product of the key and query to write into a memory matrix. To retrieve information from the memory, the query does a dot product with the memory matrix. The benefit of this attention mechanism is that the computation complexity is reduced from quadradic to linear. Experiment results show the efficiency of the proposed method. The author further proposes NAM-based LSTM and NTM, both models achieve stronger performance compared to their vanilla version while maintaining a similar computation complexity.

**Summary Of The Review:**

Overall the paper proposes a new outer product-based memory mechanism, that has an incremental novelty. The author also leverages the proposed method to extend recurrent neural networks such as LSTM and NTM. The compositional generalization results are interesting, but the detail of the experiments requires further clarification.

---

> ### Author Response · Authors · 2022-11-07
> **Response to Reviewer ek6u**
>
> First of all, we appreciate your comments and feedbacks.
>
> > 1. The NAM is not substantially different from the linear transformer and achieves lower performance compare to the linear transformer while the speedup is similar.
>
> Please see the answers to the common comments. The evaluation in Section 4 is a proof-of-concept and outperforming linear transformers is not the main objective of this work.
>
>
> > 2. The performance of LSAM and NAM TM still falls behind the SOTA models in the SCAN task.
>
> For the SCAN task, please consider that LSAM and NAM-TM are not domain-speciﬁc designs. We used the length split for the SCAN task. It is speciﬁed at the fourth paragraph of Section 5.3. For the length split, there are two SOTA methods: Chen, et al. 2020, and Nye, et a. 2020. Those two are domain-speciﬁc architectures for neuro-symbolic program tasks. The other SCAN-related models, including syntactic attention (Russin et al. 2019) show similar performance with ours. We believe SCAN length split is hard to solve without a domain-speciﬁc structure.
>
> The 'Fib' row of Table 2 demonstrates that they are capable of solving compositional generalization problems that cannot be solved by others (Transformer, LSTM, DNC, ...). We are looking for tasks other than number sequence prediction to show our methods’ computational powers.
>
> > Overall the paper proposes a new outer product-based memory mechanism, that has an incremental novelty.
>
> Up to our knowledge, our paper is the ﬁrst work to present outer-product based neural network memory mechanism for a variety of purposes, such as compositional generalization. Linear Transformer took the same outer-product based approach but only targeted the efﬁcient long-range attention problem. If you know another MANN work that took a similar approach, please let us know.
>
> Chen, Xinyun, et al. "Compositional generalization via neural-symbolic stack machines." Advances in Neural Information Processing Systems 33 (2020): 1690-1701.
>
> Nye, Maxwell, et al. "Learning compositional rules via neural program synthesis." Advances in Neural Information Processing Systems 33 (2020): 10832-10842.
>
> Russin, Jake, et al. "Compositional generalization in a deep seq2seq model by separating syntax and semantics." arXiv preprint arXiv:1904.09708 (2019).

---

> > ### Comment · Reviewer_ek6u · 2022-11-14
> > **Fast weights is using outer-product-based memory mechanism**
> >
> > Thanks for your clarification. Please note that the Fast weights [1] are also using an outer-product-based memory mechanism.
> > [1] Ba, J., Hinton, G. E., Mnih, V., Leibo, J. Z., & Ionescu, C. (2016). Using fast weights to attend to the recent past. Advances in neural information processing systems, 29.

---

> > > ### Author Response · Authors · 2022-11-14
> > > **Thank you for letting us know about Fast weights**
> > >
> > > We appreciate that you showed us a relevant work that have not been known to us.
> > > Fast weights do use outer product to memorize the past hidden activations so that its method is similar to NAM.
> > > NAM is a more generalized form of memory architecture that supports reading, writing, erasing, and key-based addressing mechanisms.
> > > Fast weights are limited for time-decaying associative recall, while NAM allows building flexible data structures like a Turing tape in NAM-TM.
> > > For example, NAM-TM's LEFT, RIGHT, and JUMP actions could not be possible without the key-based addressing mechanism of NAM.
> > > In Fast weights, the addressing is tied to the hidden activations so that one cannot separate positions and values like in NAM-TM.
> > > Also, the only way to erase the existing information in Fast weights is to wait until it decays off, but $WR$ operation of NAM incorporates instant erasure of information which is crucial for fully-capable memory architectures.
> > >
> > > Like Linear Transformer, Fast weights can be seen as a special variant of NAM which compromises some capabilities.
> > > Although not exactly equivalent, $A' = WR(A, h, h, \eta, (1-\lambda)) = A + \eta h h^\top - (1-\lambda) (Ah)h^\top$ will be similar to the update rule of Fast weights $A' = \lambda A + \eta h h^\top$.
> > > If we add decaying constant to the $WR$, Fast weights can be exactly formulated using NAM. Time-decaying the memory matrix is an interesting idea, so we will try to add this discussion later.

---

### Official Review · Reviewer_83W7 · 2022-10-25

**Confidence:** 5
**Correctness:** 3
**Technical Novelty And Significance:** 1
**Empirical Novelty And Significance:** 1
**Recommendation:** 3

**Clarity, Quality, Novelty And Reproducibility:**

Clarify: The presentation of the problem and proposal is clear.
Quality: The quality of this work is subpar because it is unclear why the proposed method is better than Katharopoulos et al 2020 and other alternatives.
Novelty: Very limited considering Katharopoulos et al 2020.
Reproducibility: The algorithm seems to be straightforward to implement.

**Strength And Weaknesses:**

Strength
Efficient attention is a topic of general interests to the community. The presentation of the proposed method is clear. It is shown that the method can be applied to different settings such as LSTMs and NTMs.

Weakness
1. The novelty in compression with the linear attention work (Katharopoulos et al 2020) is very limited. In fact the mathematical forms are almost the same. The difference is that this work uses "unit vector normalization" so that they do not need to compute the causal masking factor Z. It is unclear to me what benefit, if any, the unit vector normalization provides.
2. In the long-range task evaluation section, the proposed method performs worse than Katharopoulos et al 2020. Also, other linear transformer works (discussed in Section 2.1) should be compared with too.
3. One more comment: although the proposed approach is linear to the sequence length, the writing process is a sequential operation. In practice, this can be slow on GPUs/TPUs in comparison with conventional attention which can be fully parallelized. It would be nice to discuss the implementation as well as the break even sequence length with efficient GPU or TPU implementations.

**Summary Of The Paper:**

This work proposes a method called neural attention memory. It constructs the attention output leveraging a memory matrix. The computational cost is linear to the sequence length (vs. quadratic in the conventional attention mechanism). The method is evaluated for long-range sequence tasks. It is also used to derived variants of LSTM and neural tuning machine.




**Summary Of The Review:**

The proposed method is very limited in its novelty in comparison with Katharopoulos et al 2020. The empirical results do not justify the merits of the method among other alternatives.

---

> ### Author Response · Authors · 2022-11-07
> **Response to Reviewer 83W7**
>
> First of all, we appreciate your comments and feedbacks.
>
>
> >1. The novelty in compression with the linear attention work (Katharopoulos et al 2020) is very limited. In fact the mathematical forms are almost the same. The difference is that this work uses "unit vector normalization" so that they do not need to compute the causal masking factor Z. It is unclear to me what benefit, if any, the unit vector normalization provides.
>
> Please see the common answers to the reviewers. The evaluation in Section 4 is a proof-of-concept to show that NAM is an effective and efficient attention mechanism despite those differences.
>
> Section 5 is the most important part of the paper, containing most of the novel designs and the contributions. That is, LSAM and NAM-TM are not just variants of LSTM and NTM (or DNC). They incorporate the strengths of both attention mechanism and recurrent neural network. Please consider their success as memory-augmented neural networks for evaluating our paper.
> The 'Fib' row of Table 2 demonstrates that they are capable of solving compositional generalization problems that cannot be solved by others (Transformer, LSTM, DNC, ...). Though the proposed models are not built for task-specific problems, they outperform traditional models in algorithmic tasks.
>
> We would appreciate if you further review Section 5 and give comments and feedbacks about it to us.
>
>
>
>
> >2. In the long-range task evaluation section, the proposed method performs worse than Katharopoulos et al 2020. Also, other linear transformer works (discussed in Section 2.1) should be compared with too.
>
> We will add more data points using other efficient transformers to Table 1 for better analysis.
>
>
>
> >3. One more comment: although the proposed approach is linear to the sequence length, the writing process is a sequential operation. In practice, this can be slow on GPUs/TPUs in comparison with conventional attention which can be fully parallelized. It would be nice to discuss the implementation as well as the break even sequence length with efficient GPU or TPU implementations.
>
> The NAM-Transformer design is a special case of NAM whose reads and writes can be computed fully in parallel. As mentioned in Section 4.3, the speedups are evaluated on a high-end GPU (RTX 3080).

---

### Author Response · Authors · 2022-11-07
**Response to common comments and questions**

Thank you for your reviews and questions. We will upload the revision of the paper reflecting your comments soon.

The biggest difference between our paper and Katharopoulos et al. 2020 (Linear Transformer) is that ours is a generic method for implementing memory structures. While Linear Transformer specifically targets attention efficiency for long-range tasks, NAM targets more general objectives, including compositional generalization, hierarchical data modeling, and few-shot learning.

Please consider Section 4 as a proof-of-concept. NAM-Transformer is a special case of NAM whose reads and writes can be computed fully in parallel. There are a few design choices (unit vector normalization, no causal Z factor, gated read/write, ...) that may affect the efficacy of NAM as an attention mechanism. NAM-Transformer is designed to be identical to Linear Transformer except the above characteristics to show that NAM remains effective and efficient despite the design differences. Outperforming other efficient transformers in long-range tasks is not the main objective of this work.

We understand and agree that the writings of Section 3 and 4 can be confusing to readers. We are working on re-organizing those sections to clarify the relationship between NAM and NAM-Transformer. We will upload the revision before Nov 18 and let you know when it is ready.

---

### Author Response · Authors · 2022-11-15
**Uploaded the revision**

Dear Reviewers,

We have uploaded a revision of the paper.
The main changes are done in Section 3 and 4, to make the relationship between NAM and NAM-Transformer more clearly.
There are some minor changes in Abstract, Section 1, and Section 2.

---

### Decision · Program_Chairs · 2023-01-20

**Decision:**

Reject

**Justification For Why Not Higher Score:**

Written in the meta-review.

**Justification For Why Not Lower Score:**

N/A

**Metareview: Summary, Strengths And Weaknesses:**

This paper proposes a memory-based attention mechanism called neural attention memory (NAM), to avoid the quadratic bottleneck of transformers. The proposed strategy is similar to prior work with linear transformers (Katharopoulos et al. 2020): it first writes a memory matrix by adding outer products of key-value pairs, and then reads it by multiplying the memory matrix with a unit query vector. NAM attention reduces the computational complexity to linear with respect to the sequence length. Experiments are shown in long-range arena and on compositional generalization tasks like number sequence prediction, sequence reduction, and SCAN. While the paper addresses a relevant problem and the empirical results look fairly convincing, reviewers pointed out as the main weakness the insufficient novelty with respect to previous methods, in particular the linear transformer, and lack of comparison with other methods. The revised version addresses some of these points, but it is insufficient to alleviate the main concerns. I therefore recommend rejection. I recommend the authors take into account the comments of the reviewers to improve their paper in a future iteration of their work.